# High-frequency FeSiAl-based soft magnetic composites via simultaneously suppressed eddy and hysteresis losses

Hongxia Li[1,3], Yixing Li [2,3], Rongzhi Zhao [1]✉, Sateesh Bandaru[1], Zhenhua Zhang[1], Hong Pan[1], Jintao Lin[1], Zhaoyuan Liu[1], Xiaolian Liu[1], Chenglong Hu[1], Pengfei Guan [1], Anjian Pan[1], Erpan Zhang[1], Zhong Li[1], Huawei Rong[1] & Xuefeng Zhang [1]✉

Soft magnetic composites present promising solutions for integrated transformers and inductors, but challenges arise at MHz range, where hysteresis and intra-eddy losses result in substantial performance degradation. In this study, we propose a bulk/interface insulation strategy to synthesize FeSiAl:Sn/ $Al_2O_3$ soft magnetic composite by mutual diffusion of metal atoms, where a ~ 3 um-depth Sn-substituted FeSiAl is obtained in the matrix and an insulating $Al_2O_3$ layer is epitaxially grown on FeSiAl surface. The formation of FeSiAl:Sn can not only suppress intra-eddy loss by enhancing electrical resistivity, but also reduce coercivity by mitigating lattice distortion for reducing hysteresis loss. Meanwhile, the in-situ grown $Al_2O_3$ layer can reduce inter-eddy loss by electrical isolation between FeSiAl particles. This construction leads to power loss of 47 mW/cm³ at 100 kHz and 1344 mW/cm³ at 1 MHz under 50 mT, as well as effective permeability of 60 up to tens of MHz, associated with cut-off frequency of 250.7 MHz. This approach lays the groundwork for the development of high-frequency soft magnetic composites in engineering applications.

The rapid advancement of emerging industries, including new energy vehicles, photovoltaic technologies, and 5 G/6 G communications, has led to an urgent demand for high-performance soft magnetic composites (SMCs) to meet increasing requirements[1–4]. SMCs are characterized by their ability to rapidly magnetize and demagnetize in response to an external magnetic field, thereby facilitating energy storage and conversion while suppressing noise in electrical machines and power electronics[5,6]. However, it is noteworthy that over 9% of electric energy is dissipated as heat during transmission and distribution, primarily due to power loss ($P_{cv}$) in SMCs[7]. This loss predominantly arises from hysteresis loss ($P_h$) and eddy loss ($P_e$), with eddy loss further categorized into intra- and inter-part components[8,9]. To mitigate power loss, much research effort has been dedicated to

developing insulating techniques to achieve electrical isolation by employing interfacial insulating layers[10–15].

Various insulating techniques can be generally classified into in-situ insulation (such as acid-base passivation, solid-phase reaction, etc.) and ex-situ insulation[16–20]. Compared with the ex-situ insulation, the in-situ $Al_2O_3$ layer is obtained through the in-situ chemical reaction of Al atoms by harnessing segregation in the matrix, exhibiting better lattice match and stronger chemical bonding with the matrix. It is less likely to crack or fall off during subsequent molding process, which is conducive to achieving homogenous insulation layer between particles and effectively reducing eddy current loss. However, the segregation of Al atoms in the matrix will introduce vacancies in the matrix, causing lattice shrinkage and increasing anisotropy, which

[1]Institute of Advanced Magnetic Materials, College of Materials and Environmental Engineering, Hangzhou Dianzi University, Hangzhou 310018, China. [2]Key Laboratory for Anisotropy and Texture of Materials (MOE), School of Materials Science and Engineering, Northeastern University, Shenyang 110819, China. [3]These authors contributed equally: Hongxia Li, Yixing Li. ✉e-mail: zhaorz@hdu.edu.cn; zhang@hdu.edu.cn

deteriorates coercivity and hysteresis loss. Currently, the in-situ insulation techniques have established a significant presence in the design of kHz power devices. However, as the operational frequencies of FeSiAl SMC devices escalate into MHz, high power losses still considerably impede their application potential.

The power loss of insulated SMCs primarily stems from intra-eddy loss and hysteresis loss, where insulating techniques have limited efficacy. The intra-eddy loss reflects electrical resistivity of matrix, which can be enhanced by heterogeneous doping[21–23]. Consequently, implementing heterogeneous element doping in master alloys emerges as a promising strategy to improve electrical resistivity and inhibit power loss. For instance, Wang et al. demonstrated an enhanced electrical resistivity by doping Ti in Fe-6.5Si alloy[24]. Liu prepared $Fe_{85-x}Si_{9.6}Al_{5.4}Sn_x$ powders by induction melting and gas atomization and the corresponding SMC exhibited the lowest core loss of 124.4 mW/cm$^3$ ascribed to the addition of Sn in Fe sites[25]. Similar findings have been reported in $Fe_{85-x}Si_{9.6}Al_{5.4}Ti_x$ alloy, wherein power loss was effectively reduced to approximately 140.6 mW/cm$^3$ under the same conditions[26]. Nevertheless, unexpected diffusion and segregation happen for the doping alloy during following gas atomization and annealing processes, leading to lattice distortion and influencing magneto-crystalline anisotropy[27], which in turn degrades coercivity ($H_c$) and hysteresis loss in SMCs. Furthermore, the massive heterogeneous doping in alloy can induce significant grain refinement and pinning effects at grain boundaries, further increasing hysteresis loss[25,28]. Therefore, simultaneously suppressing intra-eddy loss and hysteresis loss presents a critical challenge for SMCs.

In this study, we propose a bulk/interface insulation strategy to synthesize FeSiAl:Sn/Al$_2$O$_3$ SMCs by mutual diffusion of metal atoms during the annealing process. The SnO layer is applied to the surface of FeSiAl initially, which introduces metallic Sn and SnO$_2$ upon annealing. Then, the Al atoms in FeSiAl matrix are heteroatom substituted by the inward diffusion of Sn atoms and a ~ 3 μm-depth FeSiAl:Sn is obtained in the matrix, while the in-situ epitaxial Al$_2$O$_3$ insulating layer is formed by the aluminothermic reaction between Al and SnO$_2$ with the outward diffusion of substituted Al atoms. The incorporation of Sn, which has a larger atomic radius compared to Al, mitigates lattice distortion during annealing, thereby inhibiting coercivity and ultimately resulting in decreased hysteresis loss. In addition, the substituting Sn can enhance electrical resistivity of the matrix, effectively inhibiting intra-eddy loss, while the inter-eddy loss is reduced by electrical isolation of Al$_2$O$_3$ insulating layer between FeSiAl particles. Consequently, the power loss at 50 mT is effectively reduced to 47 mW/cm$^3$ and 1344 mW/cm$^3$ at 100 kHz and 1 MHz, respectively, with stable effective permeability ($\mu_e$) of 60 up to tens of MHz and cut-off frequency of 250.7 MHz.

## Results

### Composites construction

The schematic representation of bulk/interface by mutual diffusion of metal atoms to achieve FeSiAl:Sn/Al$_2$O$_3$ is depicted in Fig. 1a. In this process, SnO is generated by annealing SnCl$_2$·2H$_2$O in a nitrogen atmosphere on the surface of FeSiAl particles. This condition initiates a disproportionation reaction, yielding both SnO$_2$ and metallic Sn[29]. Consequently, the Sn atoms diffuse inward and gradually substitute Al in the FeSiAl matrix[30], while outward diffusion of Al atoms forming an Al-rich layer at the particle surface. As Al and SnO$_2$ accumulate, an aluminothermic reaction is triggered, leading to the in-situ formation of an epitaxial Al$_2$O$_3$ insulating layer and thereby realizing the structure of FeSiAl:Sn/Al$_2$O$_3$[31–33]. This Al$_2$O$_3$ insulating layer reduces inter-eddy loss attributed to electrical isolation between the FeSiAl particles (Fig. 1b). Furthermore, the substituting Sn heteroatoms can release lattice distortion during annealing treatment, resulting in reduced coercivity and mitigating hysteresis loss. Additionally, the Sn substitutions in the FeSiAl matrix enhance its electrical resistivity, contributing to a decrease in intra-eddy loss (Fig. 1c).

To elucidate the underlying construction process, we employed a series of experimental characterizations and theoretical calculations. Initially, the occurrence of the disproportionation reaction was confirmed through X-ray diffraction (XRD), thermogravimetric analysis (TGA), and transmission electron microscopy (TEM) as depicted in Supplementary Fig. 1. The results indicate the formation of Sn and SnO$_2$ following the annealing of SnCl$_2$·2H$_2$O. Subsequently, various substitution configurations were established: mono-substitution of Fe with Sn (Sn@Fe), Si with Sn (Sn@Si), and Al with Sn (Sn@Al), as well as bi-substitutions (2Sn@2Al) and tri-substitutions (3Sn@3Al). The most stable configurations of FeSiAl, Sn@Fe, Sn@Si, and Sn@Al are presented in Supplementary Fig. 2a. Notably, all formation energies for the Sn@Al configurations are found to be negative, in contrast to the positive formation energies of the other configurations (Supplementary Fig. 2b). This observation indicates that Al atoms exhibit a higher propensity for substitution by Sn atoms than Fe and Si atoms. Moreover, the formation energies of the 2Sn@2Al and 3Sn@3Al configurations were calculated at −0.017 eV and −0.025 eV, respectively, suggesting relatively low formation barriers for Sn substitution Al within the FeSiAl matrix.

In the context of the aluminothermic reaction, we examined variations in Gibbs free energy based on simulation models as illustrated in Supplementary Fig. 3a, b. Notably, the Gibbs free energy per unit of SnO$_2$ is significantly higher than that of Al$_2$O$_3$ (Supplementary Fig. 3c). The Gibbs free energy decreases from −22.58 to −41.23 eV during the transformation from Al+SnO$_2$ to Sn+Al$_2$O$_3$, providing strong support for the formation of Al$_2$O$_3$ and metallic Sn at the surface of FeSiAl through the aluminothermic reaction. To assess the phase composition of FeSiAl at various stages, we analyzed the XRD patterns of FeSiAl, annealed FeSiAl and FeSiAl:Sn particles as depicted in Supplementary Fig. 4a. Compared with FeSiAl, the annealed FeSiAl and FeSiAl:Sn samples reveal a complete DO3 phase (JCPDS: 451206)[34], verified by electron backscatter diffraction (EBSD) results shown in Supplementary Fig. 4b, c. A right shift of the (220) peak to 44.963° in the annealed FeSiAl is observed ascribed to segregation of Al atoms in Supplementary Fig. 4d. However, the (220) peak has shifted back to 44.939° in the annealed FeSiAl:Sn, suggesting that lattice shrinkage is released due to the incorporation of Sn within the FeSiAl matrix. Furthermore, cross-sectional scanning electron microscopy (SEM) images, combined with energy dispersive spectroscopy (EDS) mapping of the FeSiAl:Sn SMC, provide additional insights into the heteroatom substitution and aluminothermic reaction (Supplementary Fig. 5). A distribution of Sn within the FeSiAl matrix and a distinct oxide layer on the FeSiAl particles are clearly detectable, indicating successful substitution and formation mechanisms.

### Microstructure characterizations

To evaluate the microstructures of the FeSiAl:Sn/Al$_2$O$_3$, their interfacial structures have been characterized via transmission electron microscopy (TEM) as illustrated in Fig. 2. The selected area electron diffraction (SAED) pattern in Fig. 2a, reveals diffraction spots corresponding to the $(0\bar{2}2)$ and $(2\bar{2}0)$ planes, indicative of the DO3 structure in FeSiAl. Figure 2b, c depicts the interfacial structure and elemental distribution within the FeSiAl:Sn-0.8 SMC, where Sn-rich and Al$_2$O$_3$ layers are clearly observed encasing the FeSiAl matrix. Elemental distribution, extracted along the cyan arrow in Fig. 2c and summarized in Fig. 2d, further substantiates the FeSiAl:Sn/Al$_2$O$_3$ architecture as evidenced by the varying intensities of Al, Sn, and O elements with respect to position. Additionally, high-angle annular dark-field (HAADF) mapping and line scanning results (Supplementary Fig. 6) also corroborate these findings.

To explore the lattice structure of FeSiAl:Sn/Al$_2$O$_3$, we obtained high-resolution TEM (HRTEM) images of the region highlighted in the cyan box of Fig. 2b as shown in Fig. 2e. The lattice fringes corresponding to the $(\bar{3}11)$ and (222) planes with a zone axis of $[0\bar{1}1]$ in

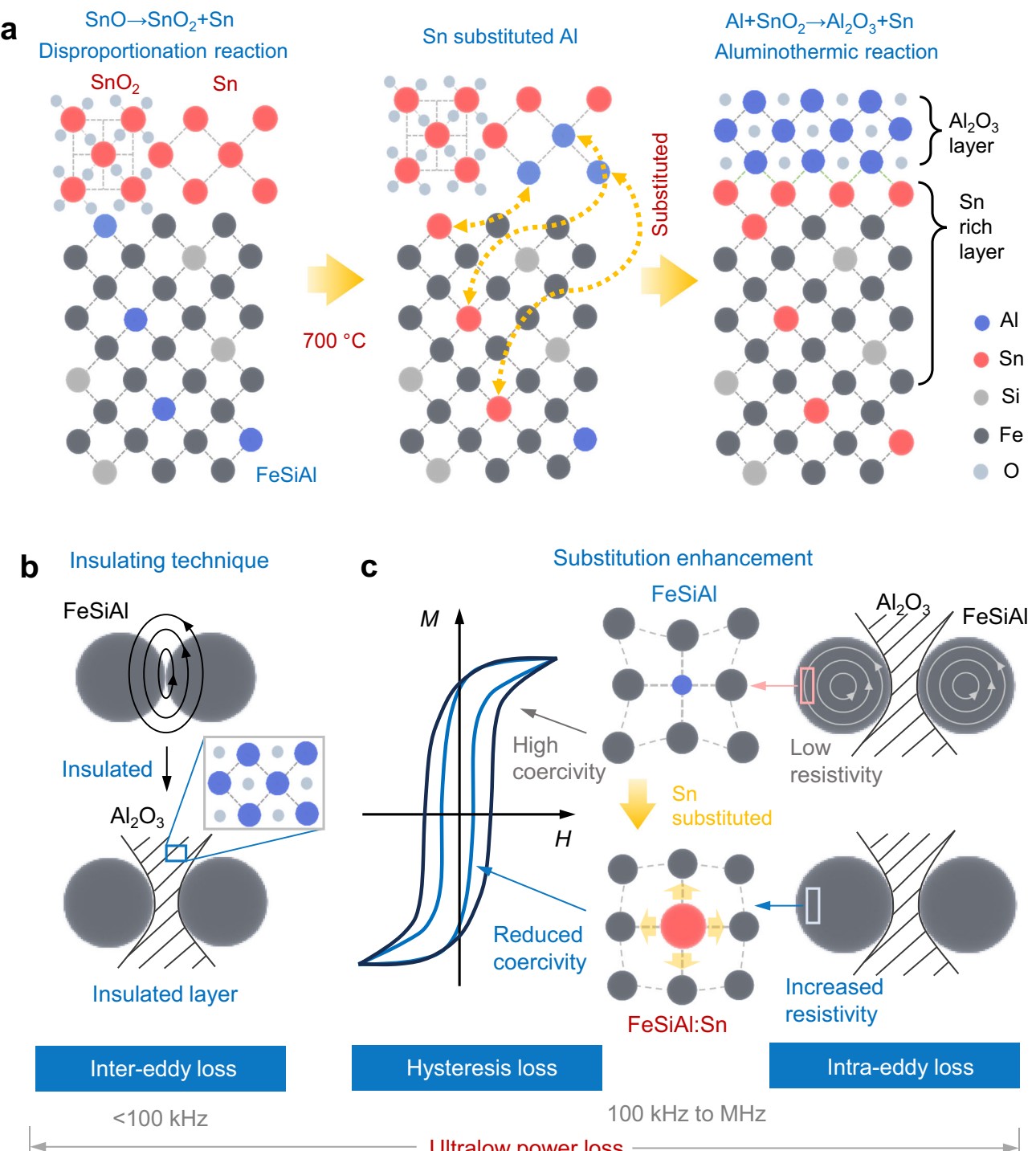

**Fig. 1 | Origination of FeSiAl:Sn/Al$_2$O$_3$ structure. a** Schematic diagram of the substitution of Sn and formation of Al$_2$O$_3$ by mutual diffusion. **b, c** Schematic representation of the power loss suppression mechanism in FeSiAl:Sn SMCs. Inter-eddy loss is effectively mitigated through the incorporation of an insulating Al$_2$O$_3$ layer (**b**). In parallel, hysteresis and intra-eddy losses are suppressed by substituting Sn for Al, which results in a reduction of coercivity and an increase in electrical resistivity (**c**).

Fig. 2e1, confirm the presence of Al$_2$O$_3$ (JCPDS:100425). Conversely, the (02$\bar{2}$) and (20$\bar{2}$) planes in the FeSiAl and Sn-rich layer correspond to the lattice fringes of the DO3 structure (Fig. 2e2, e3). Notably, the epitaxial growth of the Al$_2$O$_3$ layer is demonstrated by the aligned ($\bar{3}$11) plane of Al$_2$O$_3$ and the (20$\bar{2}$) plane of the Sn-rich layer. To investigate the substitution depth of Sn within the FeSiAl matrix, HAADF and EDS mapping images were conducted as shown in Supplementary Fig. 7a-c. Results indicate a broad distribution of Sn atoms (about 3 μm)

throughout the FeSiAl matrix with concentration decreasing gradually along the substitution depth (Fig. 3a, b).

High-angle annular dark-field scanning transmission electron microscopy (HAADF-STEM) was employed to elucidate the distribution of substituting Sn atoms. Figure 3c illustrates atomic columns of Fe and Si/Al viewed along the [110] direction, revealing periodic arrangements of three golden Fe atom columns and one Si/Al atom column. Although thickness contrast from multiple atomic columns

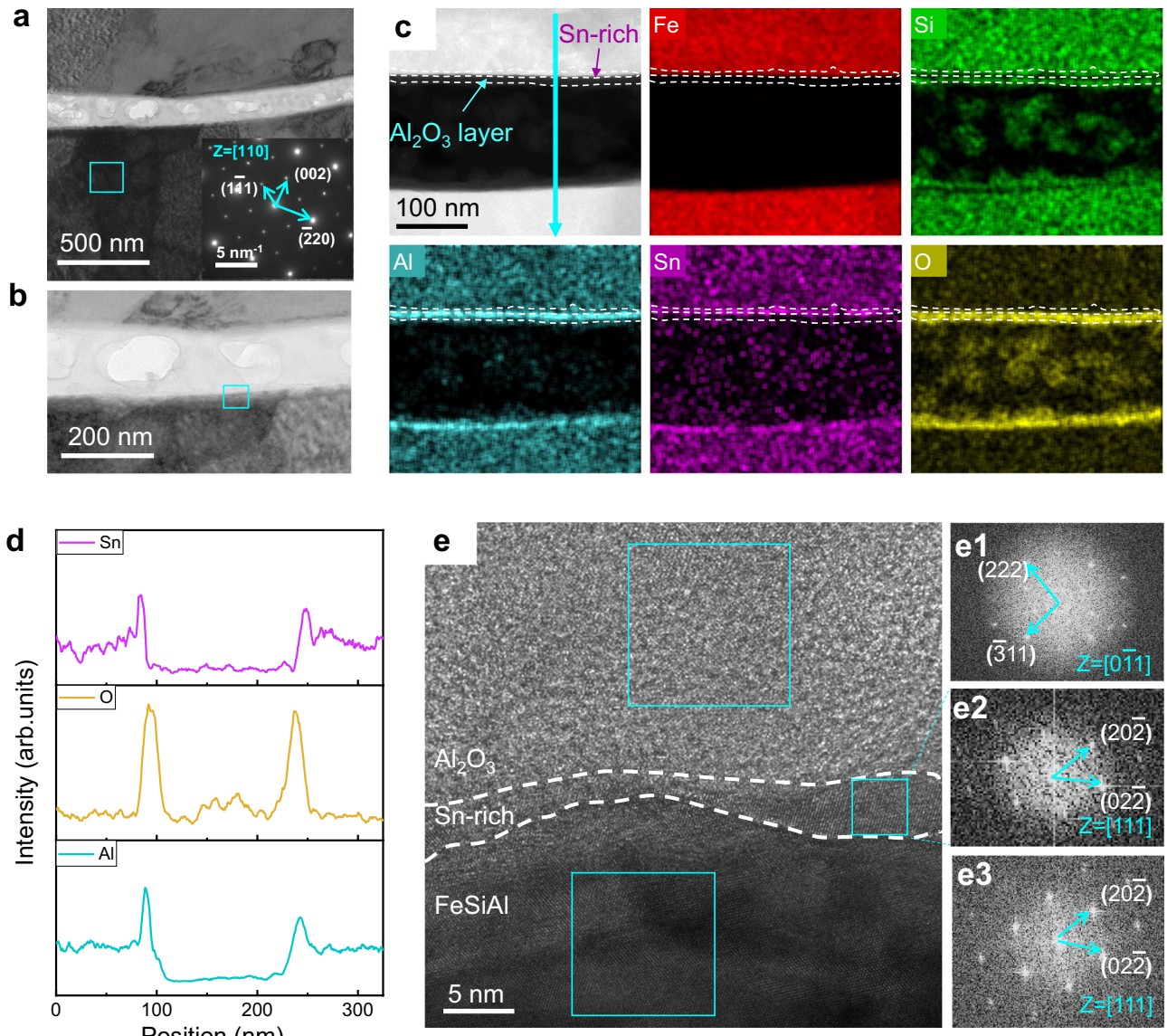

**Fig. 2 | Microstructure characterization. a, b** TEM image of FeSiAl:Sn-0.8 SMC. The inset corresponds to the SAED of the cyan box. **c** HAADF image and corresponding EDS mapping images of FeSiAl:Sn SMC. **d** The Sn, O, and Al intensity profiles along the cyan arrow in **c**. **e** HRTEM image of the cyan box in **b**. **e1-e3** Fast Fourier transform (FFT) images of the corresponding cyan box in e. Source data are provided as a Source Data file.

may be present, the primary contributor to atomic brightness is the atomic number contrast (Z contrast). The atomic-scale microstructure of the FeSiAl:Sn matrix in Fig. 3d-f, clearly differentiates between bright Fe atom columns and darker Si/Al atom columns due to the substantial atomic number difference between Fe and Si/Al. Consequently, a brighter profile at the Si/Al sites (position I, Fig. 3g) is attributed to Sn atoms (atomic number: 50) substituting Al atoms (atomic number: 13). Medium and weak contrast profiles for Si/Al columns at positions II and III (Fig. 3h-i) are also observed, indicating lower concentrations of Sn substitution Al.

To enhance contrast, normalized atomic-resolution HAADF-STEM images were calculated, which facilitate clearer visual identification, given the direct proportionality between HAADF contrast and the mean atomic number ($Z^2_{mean}$)[35,36]. The colors of the circles denote atomic intensity contrast at various Si/Al column sites: red circles with strong atomic intensity contrast at position I near the interface (Fig. 3j), pink circles indicating medium atomic intensity at position II (Fig. 3k), and blue circles denoting low atomic intensity at position III (Fig. 3l).

This color coding clearly reflects the gradient distribution of Sn substitution Al along the depth profile. HAADF-STEM images and corresponding fast Fourier transform (FFT) analyses presented in Supplementary Fig. 8 confirm the face-centered cubic (FCC) structure of the FeSiAl:Sn matrix across different positions with a zone axis of [110]. Additionally, the interplanar spacing of the FeSiAl:Sn matrix exhibits a slight increase from position III to position I, reinforcing the gradient distribution of Sn substitution Al along the depth of the matrix. Line scan and normalized area atomic intensity profiles of Fe and Si/Al atomic columns in Supplementary Fig. 9 further validate these observations.

## Enhancement of electrical resistivity

To elucidate the electrical properties of FeSiAl-based samples, height images and surface electric potential of FeSiAl:Sn-0.8 SMC have been characterized by a Kelvin probe force microscope (KPFM) as shown in Fig. 4a, b. A homogeneous and dense $Al_2O_3$ insulating layer is observed on the surface of the FeSiAl particles, where the surface electric

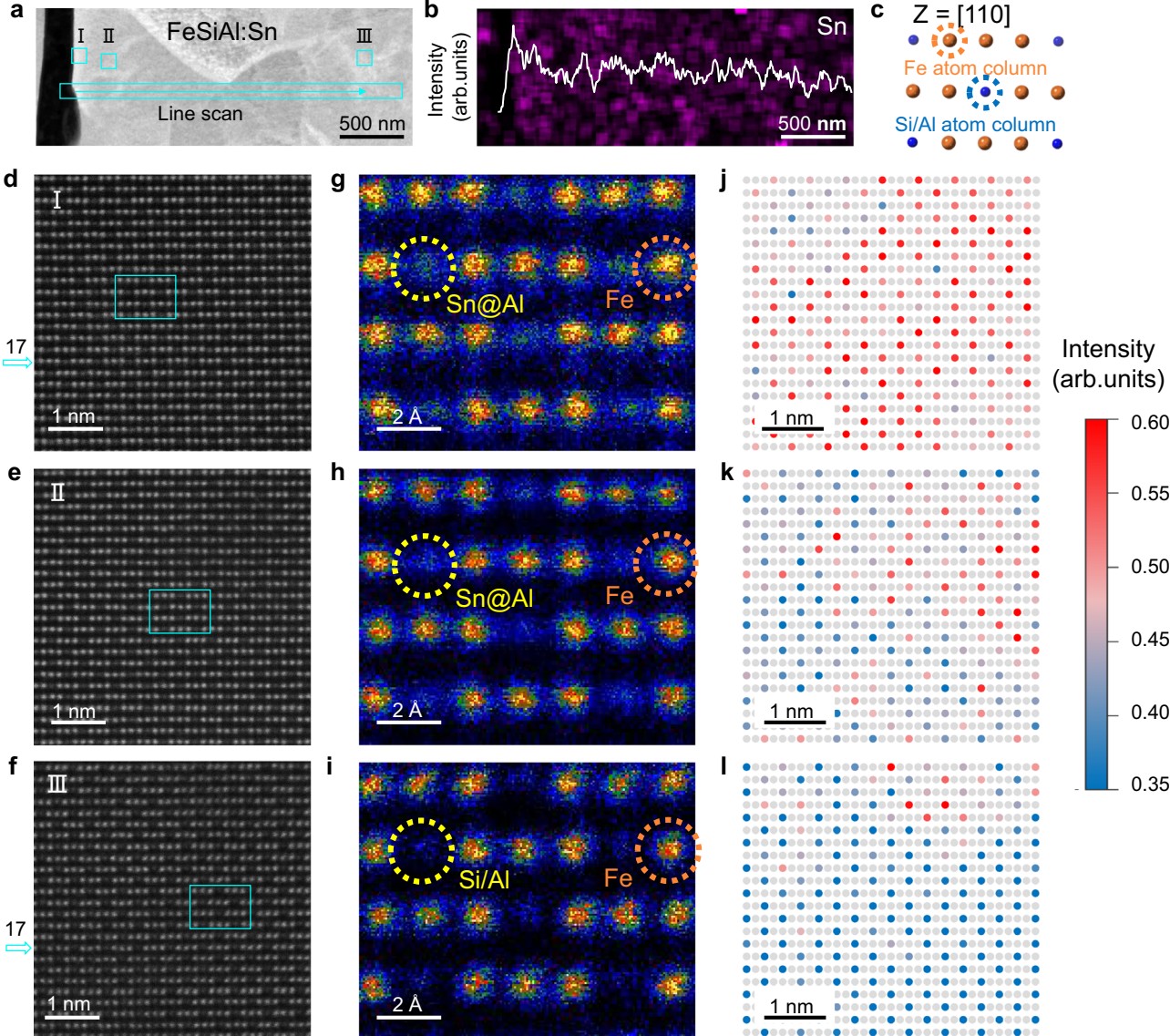

**Fig. 3 | Gradient distribution of Sn substitution Al in FeSiAl matrix. a** HAADF image of FeSiAl:Sn matrix. **b** Sn intensity profile along the cyan line scan in **a**. **c** Fe, Si/Al atom columns viewed along [110] direction, where the blue circles stand for Al and Si atom columns, and the golden circles stand for Fe atom columns. **d-f** Atomic-resolution HAADF-STEM images of FeSiAl:Sn matrix corresponding to different positions in **a** with zone axis of [110]. **g-i** The enlarged images of the cyan box in **d**, **e**, and **f**, respectively. **j-l** Normalized atomic-resolution HAADF-STEM images corresponding to **d**, **e**, and **f**, respectively. Light grey circles represent Fe atom columns, while the other color of circles indicates the atomic intensity contrast of different Si/Al atom columns. Source data are provided as a Source Data file.

potential of this insulating layer is significantly lower than that of the FeSiAl matrix. This $Al_2O_3$ insulating layer is critical for ensuring effective electrical isolation between the FeSiAl particles, thereby leading to reduced inter-eddy loss.

Electrical resistivity measurements of annealed FeSiAl:Sn-0.8 particles reveal a value of 81.8 kΩ·cm (Fig. 4c), which is markedly higher than the resistivity of FeSiAl:Sn-0 particles measured at 0.064 Ω·cm. Supplementary Fig. 10 provides further insight, showing that the average electrical resistivities of different SMCs are 41.8, 98.8, 148.2, 186.8, and 190 kΩ·cm for FeSiAl:Sn-0, 0.4, 0.6, 0.8, and 1.0, respectively. These increases in electrical resistivity can primarily be attributed to the formation of the $Al_2O_3$ insulating layer and the substitution of Sn heteroatoms within the FeSiAl matrix. Figure 4d, e depicts the height images and surface current measurements of FeSiAl particles within the FeSiAl:Sn-0 SMC. The measured surface current of FeSiAl:Sn-0 SMC is recorded at 12.95 pA (Fig. 4f). In contrast, the current value in the FeSiAl:Sn-0.8 SMC is significantly reduced to 4.29 pA (Fig. 4g-i), attributed to the increased electrical resistivity of the FeSiAl:Sn matrix, which contributes to inhibit intra-eddy loss.

## Optimal static and high-frequency magnetic properties

As previously discussed, the incorporation of Sn substitution Al mitigates lattice shrinkage compared to annealed FeSiAl particles. Geometric phase analysis (GPA) strain maps (Supplementary Fig. 11) demonstrate a uniform strain field distribution around the positions of Sn substitution Al, indicating negligible lattice distortion within the FeSiAl:Sn matrix attributable to trace amounts of Sn substitution Al. However, it is observed that lattice spacing (d-spacing) and cell volume can further increase with higher Sn substitution content. Calculations presented in Supplementary Fig. 12 suggest significant lattice distortion for structures with 2Sn@2Al and 3Sn@3Al in comparison with Sn@Al, resulting in high magneto-crystalline anisotropy energy (MAE) (Supplementary Fig. 13).

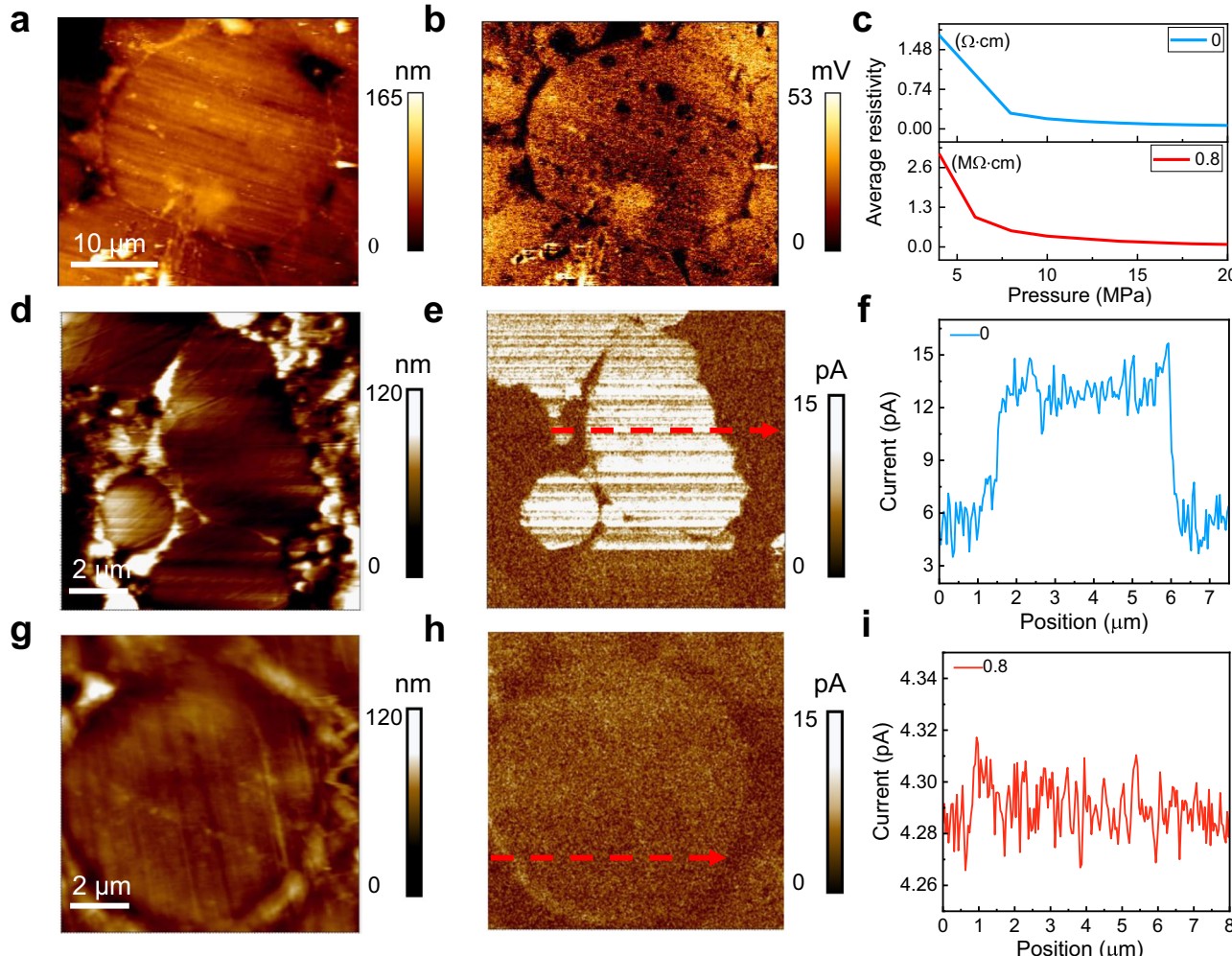

**Fig. 4 | Electrical resistivity. a, b** The height images and surface electric potential of FeSiAl:Sn-0.8 SMC. **c** Electrical resistivity of FeSiAl:Sn-0 and FeSiAl:Sn-0.8 particles after annealing. **d, e** The height images and surface current of FeSiAl particle in FeSiAl:Sn-0 SMC. **f** The corresponding current profile along the red arrow in **e. g,** **h** Height image and current map of FeSiAl:Sn particle in FeSiAl:Sn-0.8 SMC. **i** The corresponding current profile along the red arrow in h. Source data are provided as a Source Data file.

To determine the optimal amount of Sn substitution, various FeSiAl:Sn SMCs have been synthesized. As illustrated in Fig. 5a, the coercivity of these SMCs is measured from direct current (DC) hysteresis loops. Notably, FeSiAl:Sn-0.8 exhibits the lowest average coercivity of 26.59 A/m, consistent with the lowest coercive field measured through dynamic B-H assessments (Fig. 5a). It is worth noting that further increase in Sn substitution deteriorates the coercivity. Given that coercivity is generally related to magneto-crystalline anisotropy[37], the reduced coercivity of FeSiAl:Sn-0.8 confirms the decline of anisotropy, which is in agreement with the MAE simulation. By performing micromagnetic simulations, it is also found that the coercivity decreases with optimal Sn doping induced by the decrease of magneto-crystalline anisotropy (Fig. 5b), which can mitigate hysteresis loss.

Power loss, a critical performance metric for SMCs, was characterized across all samples. The power loss for FeSiAl:Sn-0 is measured at 2573 mW/cm$^3$ at 1 MHz under 50 mT, while all FeSiAl:Sn SMCs exhibit substantially lower power loss (Fig. 5c). Specifically, FeSiAl:Sn-0.8 achieves the lowest power loss of 1344 mW/cm$^3$, primarily ascribed to suppressed hysteresis and eddy losses, which is also verified by the loss separation results in Supplementary Fig. 14 and Table S1. It is noteworthy that FeSiAl:Sn-1.0 displays a relatively higher power loss of 1918 mW/cm$^3$ due to increased hysteresis loss confirmed by high

coercivity, which is mainly caused by large lattice distortion ascribed to excessive Sn substitution and increased demagnetization field due to the increase of air gap indued by the porosity. Meanwhile, FeSiAl:Sn-0.8 exhibits the highest quality factor (Q) as indicated in Supplementary Fig. 15. It should be noted that the reduced inter and intra-eddy losses of FeSiAl:Sn/Al$_2$O$_3$ structure are also verified by the inhibited current density cloud and corresponding vector maps in Supplementary Fig. 16.

Additionally, FeSiAl:Sn-0.8 SMC demonstrates a steady effective permeability of 60 across the frequency range of 0 to 1MHz (Fig. 5d). The effective permeability decreases with increasing SnCl$_2$ content, following the same trend as density. In addition to magnetic dilution due to the addition of SnCl$_2$ to the SMC, higher porosity evidenced by lower density also introduces air gaps and enlarges demagnetization field in the SMC, thereby reducing the effective permeability of FeSiAl:Sn-1.0 SMC. Meanwhile, the saturation magnetizations of FeSiAl:Sn particles decrease from 144 to 136 emu/g (Supplementary Fig. 17a). Furthermore, the stability of effective permeability in FeSiAl:Sn-0.8 across varying temperatures is also evaluated (Supplementary Fig. 17b), revealing stable effective permeability from 0 to 140 °C, which underscores its significant application potential. Permeability at 1MHz and cut-off frequencies for all FeSiAl:Sn SMCs in Fig. 5e, illustrates the effective permeability of 60 up to tens of MHz. Meanwhile,

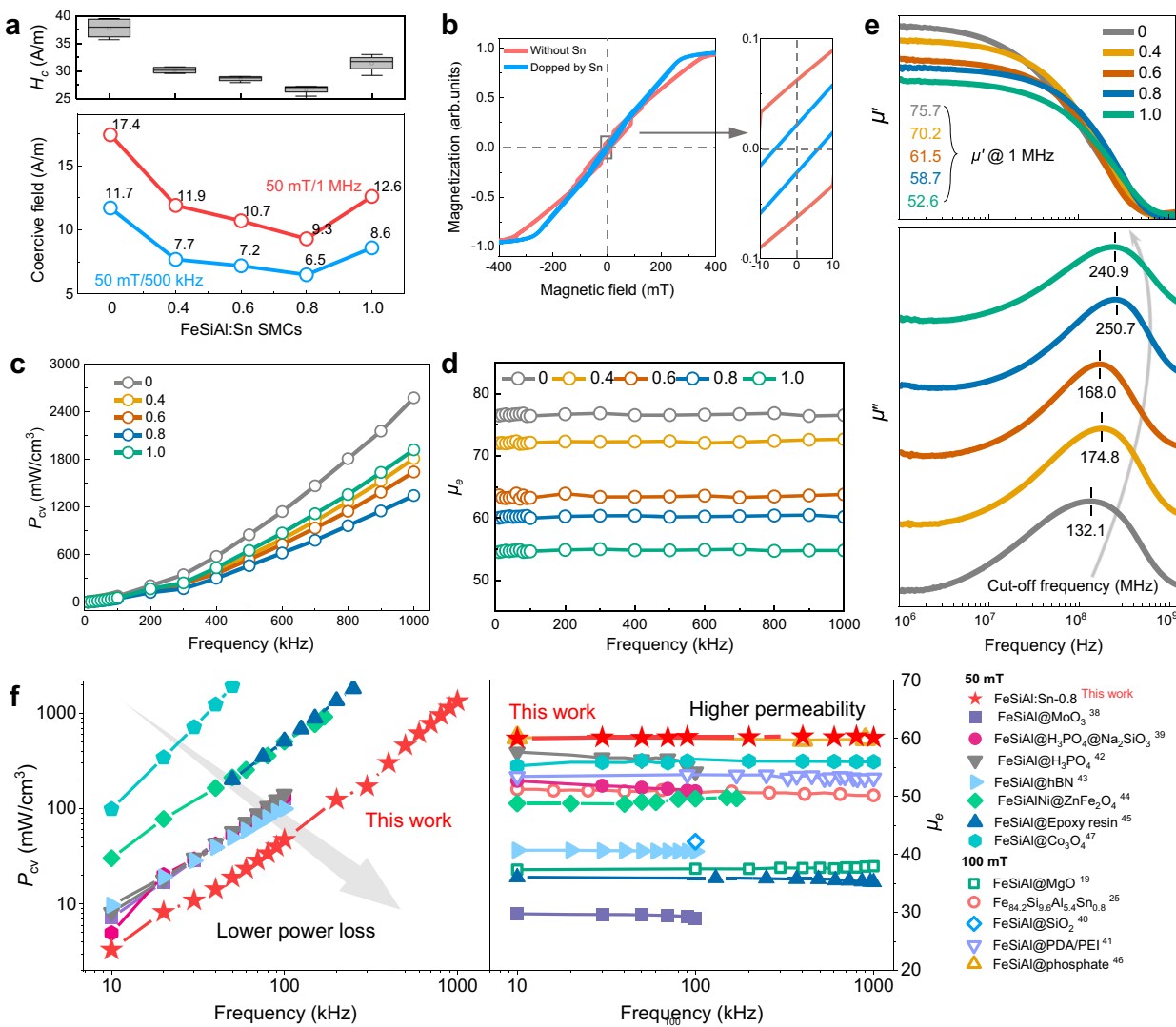

**Fig. 5 | Static and dynamic magnetic properties. a** Coercivity ($H_c$) of different FeSiAl:Sn SMCs from DC hysteresis loop and coercive field of different FeSiAl:Sn SMCs from B-H analyzer. **b** Simulated hysteresis loop of FeSiAl with and without Sn doping, in which the coercivity has been reduced with Sn doping. **c** Power loss ($P_{cv}$) of different FeSiAl:Sn SMCs as a function of frequency. The values are measured at 50 mT from a B-H analyzer. **d** Effective permeability ($\mu_e$) of different FeSiAl:Sn SMCs as a function of frequency. **e** Frequency dependence of real part of permeability $\mu'$, imaginary part of permeability $\mu''$ of different FeSiAl:Sn SMCs, including the $\mu'$@1 MHz and the cut-off frequencies of all the samples. **f** Comparison of power loss and effective permeability of our FeSiAl:Sn-0.8 SMC with other counterparts. Source data are provided as a Source Data file.

the cut-off frequency for FeSiAl:Sn-0.8 reaches to 250.7 MHz, nearly double that of FeSiAl:Sn-0 SMC (132 MHz). To visually compare the optimized performance of FeSiAl:Sn-0.8 SMC, the relationship between power loss and effective permeability as a function of frequency is presented alongside other reported counterparts in Fig. 5f [38–47] and Supplementary Fig. 18, illustrating superior magnetic performance of the FeSiAl:Sn-0.8 SMC especially in high-frequency. Additionally, the corresponding power loss values measured at specific maximum magnetic flux density and frequency are also given in Table.S2, fully confirming this point.

## Discussion

The trade-off between eddy loss and hysteresis loss significantly impacts the power loss of SMCs in high-frequency. Achieving superior performance in FeSiAl SMCs remains a challenge, particularly in reducing hysteresis loss and intra-eddy loss, despite advancements in minimizing inter-eddy loss. Key to enhancing magnetic performance and high-frequency stability is increasing the electrical resistivity of

SMCs while reducing the coercivity. Historically, the enhancement of electrical resistivity in SMCs has been limited to the introduction of insulating layers, with little exploration into improving the electrical resistivity of the FeSiAl matrix itself and simultaneously reducing coercivity to mitigate these limitations.

Our findings present a straightforward and effective solution to these challenges through the development of FeSiAl:Sn/$Al_2O_3$ composites, facilitating ultra-low power loss and maintaining frequency-stable permeability up to tens of MHz. This bulk/interface approach leverages heterogeneous substitution and encapsulated insulating layer by mutual diffusion of metal atoms, wherein a gradient distribution of Sn substitutes Al atoms in the FeSiAl matrix. This substitution effectively reduces coercivity and consequently leading to diminished hysteresis loss, which has been reconstructed by simulations. Moreover, the integration of Sn substitution Al within the FeSiAl matrix significantly enhances its electrical resistivity, directly lowering intra-eddy loss. The in-situ formation of an epitaxial $Al_2O_3$ insulating layer further suppresses inter-eddy loss and ensures stability at high

frequencies. Consequently, the FeSiAl:Sn-0.8 SMC demonstrates remarkably low power loss of 47 mW/cm$^3$ at 100 kHz and 1344 mW/cm$^3$ at 1 MHz under 50 mT, accompanied by impressive cut-off frequency of 250.7 MHz, while achieving effective permeability of 60 across the range of 0 to tens of MHz.

In summary, our work not only elucidates a methodology for enhancing high-frequency performance in FeSiAl SMCs but also offers insights applicable to the development of other material systems, potentially guided by the principles of heteroatom substitution and atom diffusion. This study lays the groundwork for future investigations aimed at optimizing materials with tailored functional layers for a variety of advanced applications.

# Methods
## Materials preparation
0.8 g Tin chloride dihydrate (SnCl$_2$·2H$_2$O) was dissolved in 20 ml absolute ethanol, and the above solution was magnetically stirred for 30 minutes with 650 rpm at room temperature to obtain a transparent solution. Then, the above solution was evenly dispersed into 500 g FeSiAl particles (TIJO, 400 mesh with D10 = 5.58 μm, D50 = 15.64 μm, D90 = 33.47 μm). FeSi$_{9.1}$Al$_{5.4}$ (wt.%) with apparent density:3.83 g/cm$^3$) and mechanically stirred for 10 minutes, so that the solution was evenly coated on the surface of FeSiAl. Then, the mixed composite was heated at 80 °C for 20 minutes (termed as FeSiAl:Sn particles). The above FeSiAl:Sn particles were mechanically mixed with 5 g silicone resin dissolved in 15 ml acetone and dried at 80 °C for 10 minutes. After adding 5 g zinc stearate, the powder was molded into toroidal cores (12.7 mm × 7.6 mm × 3.2 mm) under 1800 MPa. The compact was annealed at 700 °C for 60 minutes in N$_2$. The prepared sample was labeled as FeSiAl:Sn-0.8 SMC. Meanwhile, different FeSiAl:Sn SMCs were also prepared by controlling the amount of SnCl$_2$·2H$_2$O and absolute ethanol, which were labeled as FeSiAl:Sn-0, FeSiAl:Sn-0.4, FeSiAl:Sn-0.6 and FeSiAl:Sn-1.0 SMCs, respectively. Density of different SMCs is 6.02 g/cm$^3$, 6.01 g/cm$^3$, 6.01 g/cm$^3$, 6.0 g/cm$^3$, and 5.98 g/cm$^3$ for FeSiAl:Sn-0, 0.4, 0.6, 0.8 and 1.0, respectively.

## Microstructural characterization
The phase composition was examined by X-ray diffraction (XRD, SmartLab, Rigaku) with Cu K$_\alpha$ radiation (40 kV, 100 mA). Thermogravimetric analysis (TGA, SDT650, Taber) was used to confirm the chemical transformation during the annealing process. Electron back scatter diffraction (EBSD, SymmetryS 2, Oxford) was measured to investigate the phase composition. Surface morphology was analyzed by scanning electron microscopy (SEM, JSM-1T500HR, JEOL) with energy dispersive spectroscopy (EDS). The microstructure was observed by transmission electron microscope (TEM, Talos F200S, FEI). The elemental distribution was characterized by a high-angle annular dark-field (HAADF) detector and corresponding EDS. Before TEM observation, the sample was fabricated by cross-section milling and lift-outs using a focused ion beam (FIB, Strata 400S, FEI). The atomic resolution images of FeSiAl and FeSiAl:Sn were also characterized by spherical aberrated TEM (JEM-ARM200F, JEOL) with an energy resolution of 0.5 eV.

## Magnetic and electrical tests
The electrical resistivity of FeSiAl:Sn particles as a function of pressure and the electrical resistivity of all the FeSiAl:Sn SMCs were examined by a dynamic four probe system (ST2722-SD) and static four probe system (ST-2258C), respectively. The potential difference was measured by Kelvin probe force microscope (KPFM, JPK nanowizard 4). Current maps across the SMCs were also tested by piezo force microscopy (PFM, NT-MDT), and the applied voltage was set to 10 V. The magnetic properties were measured by a superconducting quantum interference device (SQUID, MPMS-XL-5, Quantum Design). Effective permeability and power loss were measured using a B-H

analyzer (SY-8218, IWATSU). DC hysteresis loops were characterized by a soft magnetic properties test system (FE-2100SD) using toroidal cores. The permeability spectra were measured by an impedance analyzer (E4991A, Agilent).

## Simulation
Density functional theory (DFT) calculations were performed using spin polarized DFT implemented in the Vienna Ab initio Simulation Package (VASP). The projector augmented wave potential was employed for the ion-electron terms with the valence configurations of Fe (3d$^7$4s$^1$), Si (3s$^2$3p$^2$), Al (3s$^2$p$^1$), and Sn (5s$^2$5p$^2$) in the implementation of Kresse and Joubert[48–50]. The number of plane waves in VASP was controlled by a cut-off energy of 550 eV in our calculations. The Brillouin-zone integrations were performed using the k-point grids with $3 \times 3 \times 3$ mesh for the supercell of FeSiAl and Sn-substitution system relaxation. The cut-off energy was taken as 550 eV and structural optimization was performed until the convergence criteria of energy and force was reached to $1 \times 10^{-5}$ eV and 0.02 eV/Å, respectively.

In order to model the pure FeSiAl and Sn substitution systems related to experiments, a supercell with dimensions $2 \times 2 \times 2$ was used to simulate the effects of Sn substitution. Out of the 128 atoms presenting in the supercell, mono-Fe atom substituted with mono-Sn atom (Sn@Fe), mono-Si atom substituted with mono-Sn atom (Sn@Si), mono-Al atom substituted with mono-Sn atom (Sn@Al), bi-Al atoms substituted with bi-Sn atoms (2Sn@2Al) and tri-Al atoms substituted with tri-Sn atoms (3Sn@3Al) were created and corresponding investigations were conducted, respectively.

Micromagnetic simulations were performed by using Mumax3[51]. The model was constructed by using a cubic magnetic system qualitatively, where the length of side was 300 nm. The saturation magnetization was $8.5 \times 10^5$ A/m and the exchange constant was $13 \times 10^{-12}$ J/m. The damping constant was 0.2 and the mesh size was set as $5 \times 5 \times 5$ nm$^3$. The hysteresis loop was simulated for an anisotropy constant of $5 \times 10^4$ J/m$^3$ and $5 \times 10^2$ J/m$^3$, which simulated the sample without Sn doping and with Sn doping, respectively.

The eddy losses of three different samples were calculated using the electromagnetic simulation. The model was reconstructed based on the SEM results. The first sample was the pure FeSiAl particles, and the size of the particles ranges from 1 to 40 μm. The second sample was the FeSiAl particles with Al$_2$O$_3$ insulating layer. In the third sample, Sn was doped on the surface of the FeSiAl particle, and the thickness was about 2 μm. It was assumed that all the samples were placed inside the alternating magnetic field. The strength of the filed was 50 mT, while the frequency was 100 kHz. The distribution of the current density on the surface of different samples were obtained and compared.

## Normalization of HAADF-STEM image
To visually illustrate the Sn substitution Al, we utilized normalized analysis (INA) by lowering the undesirable influence of sample thickness and signal noise[52,53]. INA quantification involved two main steps: (1) Rescaling all the Fe and Si/Al atom column intensity by subtracting the background intensity, and (2) normalizing the Si/Al atom column intensity. We performed the normalized process with a self-developed code. The normalized Si/Al atom intensity (called as $I_{Si/Al}^{Nor}$) equaled to the ratio of the intensity of Si/Al atom column to the average intensity of two nearest Fe atom columns. Thus, the normalized intensity was described as follows:

$$I_{Si/Al}^{Nor} = \frac{I_{Si/Al}^{\max}(i,j)}{[I_{Fe}^{\max}(i-1,j) + I_{Fe}^{\max}(i+1,j)]/2} \tag{1}$$

where the indices i and j labeled the position of Si/Al atom column, $I_{Si/Al}^{\max}(i,j)$ represented the maximum intensity of Al/Si atom column, while $I_{Fe}^{\max}(i-1,j)$ and $I_{Fe}^{\max}(i+1,j)$ meant the maximum intensity of the nearest two Fe atom columns.

## Data availability

The data that support the findings of this study are provided in the main text and the Supplementary Information. Source data are provided with this paper.

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

## Acknowledgements

This work is supported by the National Science Fund for Distinguished Young Scholars (52225312 X.F.Z.) and National Natural Science Foundation of China (52271173 H.X.L., 52377022 E.P.Z., 52201202 Y.X.L., 52477021 Y.X.L.).

## Author contributions

H.L., Y.L., R.Z. and X.Z. designed the experiments. J.L. and Z.L. conducted material preparation and characterization. S.B., R.Z., H.P. and P.G. carried out the simulations. Z.Z., X.L. and E.Z. performed TEM characterization. C.H. and H.R. conducted the GPA analysis. A.P. and Z.L. conducted the current density simulation. X.Z. supervised and funded this work. H.L., Y.L., R.Z. and X.Z. wrote the initial draft and revised the manuscript. All authors participated in the discussions, contributed to improve the manuscript, and approved the submitted manuscript.

## Competing interests

The authors declare no competing interests.
