## [Transparent Peer Review file · Nature Communications]

High-frequency FeSiAl-based soft magnetic composites via simultaneously suppressed eddy and hysteresis losses

Corresponding Author: Professor Xuefeng Zhang

Version 0:

Reviewer comments:

Reviewer #1

(Remarks to the Author)

The manuscript presents an in-situ insulation strategy aimed at reducing high-frequency core loss in FeSiAl-based SMCs. The approach involves the inward diffusion of Sn atoms that substitute Al in the FeSiAl matrix, accompanied by the outward diffusion of Al atoms that form an in-situ epitaxial Al_2O_3 insulating layer. This results in the formation of a $\sim 3 \mu\text{m}$ thick FeSiAl:Sn layer, and this strategy leads to a reduction of both inter- and intra-particle eddy current losses. The experimental effort is commendable, and the study addresses an important topic relevant to high-frequency magnetic materials and their application in energy-efficient devices. The authors have tried to address most of the questions raised before. However, this does not change the core message and novelty of the work, particularly with respect to literature as well as earlier report of the authors as explained below.

The revised manuscript, while technically sound and presented with a thorough experimental study, does not reach the level of conceptual advance or broad impact expected for Nature Communications.

1- The core idea of controlling eddy current pathways through in-situ oxide formation is not entirely new, and the reported power loss (47 mW/cm^3 at 100 kHz, 50 mT) closely aligns with previously reported values (e.g., 55.3 mW/cm^3 under similar conditions, DOI: <https://doi.org/10.1007/s11664-024-11580-1>). As a result, the advancement appears to be incremental rather than transformative.

2- Several technical concerns further limit the manuscript's current suitability. First, while the authors discuss diffusion-driven compositional changes, they support this mainly through EDS area scans. A simple but effective concentration line profile would better quantify the distribution of Sn and Al across the diffusion interface and validate the underlying mechanism.

3- The reported electrical resistivity of the FeSiAl particles ($0.064 \Omega\text{-cm}$) and the exceptionally low coercivity ($<30 \text{ A/m}$) are both surprisingly high and low, respectively, compared to typical literature values. Furthermore, the presentation of hysteresis data lacks clarity: the experimental loop is relegated to the Supplementary Information, while a simulated loop is shown in the main text (Fig. 5b). This makes it difficult for readers to judge the validity of the coercivity claims. A zoomed-in view of the experimental loop, especially near the coercive field, would greatly improve the transparency of the magnetic property analysis.

4- While the manuscript repeatedly emphasizes the roles of coercivity, resistivity, and porosity in influencing total core loss through the reduction of hysteresis and eddy current loss components, no quantitative decomposition of these loss contributions is provided. Core loss separation, using either experimental or analytical methods, would significantly strengthen the manuscript's conclusions and help isolate the true contribution of the proposed mechanism.

5- Although the work presents interesting experimental observations and a potentially useful approach, the novelty and impact remain limited, and several key aspects require deeper analysis and clearer validation. Based on the current scope and level of advancement, I cannot recommend the manuscript for publication in Nature Communications and it needs to be considered elsewhere with a more focus on soft magnetic materials.

Reviewer #2

(Remarks to the Author)

Comments from Ref.

The submission describes a very in depth study of a series of FeAlSi-SnO composites intended for use as low loss inductor

materials for applications at 10s kHz to ~1 Mhz frequencies. The authors strive to convince the readership of originality of approach and performance.

1. The findings are indeed interesting and of value to the inductor materials community and for energy storage (inductors), power conditioning (filters), and power conversion (converters and transformers) applications at frequencies of 10s to 100s of kHz.

2. The engineering utility of these materials are clearly apparent especially in the design of data centers among other applications.

3. Although the authors claim applications to much higher frequencies. Although the definition of cut-off frequency embraced here is used by others, in practice this value is not near the operational frequency. Therefore, these alloys do not offer practical use at the frequencies cited and they provide very low permeabilities for most applications. Finally, characterization of sample set magnetization is incomplete. Authors do not attempt to put this type data in the context of comparison to other competitive products etc. This would factor into power handling and size factors.

4. Here are my conclusions: the work is thorough and of a very high quality; the conclusions are well-thought out and defensible; the materials and compositions are largely unoriginal (for example, the literature is replete with reports of FeAlSi alloys for inductor applications, and the use of Sn-oxides as a GB additive to disrupt eddy currents is also not new); the findings are not completely surprising nor unanticipated. Based on these comments, I believe the work does not warrant publication in Nature Comm and is better suited for other materials journals.

Reviewer #3

(Remarks to the Author)

The authors have addressed my concerns in their response and revised manuscript. I recommended it for publication in Nature Comm. as is.

Reviewer #4

(Remarks to the Author)

The authors report a promising strategy for simultaneously reducing hysteresis and dynamic losses (including both intra- and inter-particle eddy currents) in a FeSiAl-based soft magnetic composite (SMC). To my knowledge, numerous studies on FeSiAl have utilized Al₂O₃ as an insulating coating, including the paper from the same research group cited by Reviewer 1. Furthermore, the strategy of doping with heteroatoms to modify the crystal lattice, such as with Ti or Sn, has been previously explored. In this regard, while the authors draw comparisons to prior work on Sn doping, this research (doi.org/10.1016/j.jmrt.2022.04.078) is not discussed in the introduction. To better contextualize and highlight the specific advancements of the current manuscript, a more thorough discussion of this reference in the Introduction would be highly beneficial.

The novelty here lies in combining these two strategies, which have previously only been used in isolation. The clever choice of precursor has allowed to generate an in-situ insulating layer while improving the electrical resistivity and minimizing the coercivity of the ferromagnetic particle, yielding results that are not only interesting from an application standpoint but also inspirational for developing new concepts in this class of materials.

Regarding the manuscript, my concerns are as follows:

1. Reviewer 1 questioned why the Sn-1.0 sample, despite having the highest resistivity, exhibits higher power loss than the Sn-0.8 sample. The authors argue this is due to a trade-off: beyond the optimum of Sn-0.8, the increase in hysteresis loss (from higher coercivity due to excessive lattice distortion) outweighs the benefit of increased resistivity. While this explanation is plausible, it overlooks a critical confounding variable: porosity. The authors' own data show that the density of the SMCs decreases with increasing Sn content (6.02 g/cm³ for Sn-0, 6.0 g/cm³ for Sn-0.8, and 5.98 g/cm³ for Sn-1.0). Lower density implies higher porosity, which acts as an effective air gap, increasing the demagnetization field, increasing hysteresis losses, and reducing the effective permeability (μ_e). This effect is evident in Fig. 5d, where μ_e for the Sn-1.0 sample is the lowest. The authors attribute the drop in μ_e solely to "magnetic dilution," but the density data suggest porosity is a contributing factor that has not been discussed. The authors should expand their discussion on the Sn-1.0 sample's performance to include the likely contribution of its lower density and higher porosity.

2. The "Material preparation" section lacks sufficient detail to ensure the work's reproducibility. While some points were addressed in response to other reviewers, I suggest the authors clarify the following:

- The mass of the FeSiAl particles mixed with the 0.8 g of tin(II) chloride dihydrate.
- The volume of ethanol used to dissolve the tin chloride and any additional parameters (e.g., stirring speed, temperature).
- The method used to mix the FeSiAl particles with the tin chloride solution (e.g., mechanical, magnetic) and the associated parameters (e.g., speed, temperature, duration).
- The drying method and parameters (e.g., temperature, duration, atmosphere) used after mixing, if any.
- The mixing procedure for the FeSiAl particles with the 5 g of silicone resin and 5 g of zinc stearate, including the mass of the FeSiAl powder used.

(f) What happens to the silicone resin after the annealing process? Was it completely burned off?

(g) The authors only mentioned that the particles (I am assuming mean size) are 400 mesh. However, Supplementary Fig. 15 reveals a quite large distribution size for the particles. Please, inform the d10, d50, and d90 of the distribution, as it can largely affect the intra-eddy losses.

3. Part of the caption for Figure 5 refers to measurements at 100 mT, but these data are not shown in the figure itself. The referenced data appear in Supplementary Figure S17, which lacks a caption. Please correct this discrepancy to improve clarity for the reader.

Version 1:

Reviewer comments:

Reviewer #1

(Remarks to the Author)

The authors have addressed all the questions raised earlier. I would like to recommend the revised manuscript for publication as it is.

Reviewer #4

(Remarks to the Author)

The authors have addressed my concerns in their response and revised manuscript. I recommended it for publication in Nature Comm. as is.

=====**Authors' responses to the reviewers' comments**=====

Response to Reviewer 1

Reviewer #1 (Remarks to the Author):

The manuscript presents an in-situ insulation strategy aimed at reducing high-frequency core loss in FeSiAl-based SMCs. The approach involves the inward diffusion of Sn atoms that substitute Al in the FeSiAl matrix, accompanied by the outward diffusion of Al atoms that form an in-situ epitaxial Al₂O₃ insulating layer. This results in the formation of a ~3 μm thick FeSiAl:Sn layer, and this strategy leads to a reduction of both inter- and intra-particle eddy current losses. The experimental effort is commendable, and the study addresses an important topic relevant to high-frequency magnetic materials and their application in energy-efficient devices. The authors have tried to address most of the questions raised before. However, this does not change the core message and novelty of the work, particularly with respect to literature as well as earlier report of the authors as explained below.

The revised manuscript, while technically sound and presented with a thorough experimental study, does not reach the level of conceptual advance or broad impact expected for Nature Communications.

1- The core idea of controlling eddy current pathways through in-situ oxide formation is not entirely new, and the reported power loss (47 mW/cm³ at 100 kHz, 50 mT) closely aligns with previously reported values (e.g., 55.3 mW/cm³ under similar conditions, DOI: <https://doi.org/10.1007/s11664-024-11580-1>). As a result, the advancement appears to be incremental rather than transformative.

Author reply: Although the approach of suppressing eddy current pathways via in-situ oxide formation has been demonstrated to be effective, we herein propose a bulk/interface insulation strategy to synthesize FeSiAl:Sn/Al₂O₃ SMCs by mutual diffusion of metal atoms. This approach not only enables the formation of an in-situ insulating layer but also enhances electrical resistivity while reducing coercivity.

Then, the reported power loss of 55.3 mW/cm³ in Ref. R1 measured at 50 mT/100 kHz (DOI: <https://doi.org/10.1007/s11664-024-11580-1>) is higher than our power loss of 47 mW/cm³. It should be more emphasized that, besides this, our other points are also far better than the literature's results, as seen in Table R1.

Table R1. Comparison of power loss of FeSiAl:Sn-0.8 SMC with the referred reference.

SMCs	P _{cv} (mW/cm ³)						
	100 mT/ 50 kHz	100 mT/ 100 kHz	50 mT/ 100 kHz	50 mT/ 300 kHz	50 mT/ 400 kHz	50 mT/ 500 kHz	50 mT/ 1 MHz
Fe ₃ O ₄ /Al ₂ O ₃ /SiO ₂ ^[R1]	149.3	329	55.3	212.2	405	-	-
FeSiAl:Sn-0.8 ^{This work}	74	190	47	172	302	462	1344
Ratio of reduction	50.4%	42.2%	15.0%	18.9%	25.4%	-	-

[R1]. Hanyu Yao, et al, *J.Elect. Mater.* (2025) 54:2027-2037.

2- Several technical concerns further limit the manuscript's current suitability. First, while the authors discuss diffusion-driven compositional changes, they support this mainly through EDS area scans. A simple but effective concentration line profile would better quantify the distribution of Sn and Al across the diffusion interface and validate the underlying mechanism.

Author reply: We have provided the atomic concentration line profile to better quantify the distribution of Sn and Al across the diffusion interface as shown in Supplementary Fig. 7 (Supporting information). Sn-rich and broad distribution of Sn atoms throughout the FeSiAl matrix is observed with intensity decreasing gradually along the substitution depth, while Al intensity also weakens from surface to the interior, verifying the Al outward diffusion in FeSiAl matrix.

Supplementary Fig. 7. Depth profile of Sn substitution. **a**, The image of whole FIB sample of FeSiAl:Sn-0.8 SMC. **b**, HAADF and corresponding EDS mapping images of the orange box in **a**. **c**, The Sn and Al atomic concentration profiles along the orange arrow in **b**.

3- The reported electrical resistivity of the FeSiAl particles ($0.064 \Omega \cdot \text{cm}$) and the exceptionally low coercivity ($<30 \text{ A/m}$) are both surprisingly high and low, respectively, compared to typical literature values. Furthermore, the presentation of hysteresis data lacks clarity: the experimental loop is relegated to the Supplementary Information, while a simulated loop is shown in the main text (Fig. 5b). This makes it difficult for readers to judge the validity of the coercivity claims. A zoomed-in view of the experimental loop, especially near the coercive field, would greatly improve the transparency of the magnetic property analysis.

Author reply: Different measuring methods could induce the discrepancy in obtained testing values. Herein, the electrical resistivity is measured by dynamic four probe system (Suzhou Lattice Electronics Co., Ltd, ST2722-SD) for FeSiAl particles, which is suitable for powder samples. We have revised the description to avoid the confusion

in the section of *Magnetic and electrical tests*. This method provides a sequence of electrical resistivity under progressively increasing applied pressure on the powders as shown in Fig. 4c of the main text. $0.064 \Omega \cdot \text{cm}$ is the electrical resistivity of FeSiAl particles under 20 MPa. Meanwhile, FeSiAl:Sn-0.8 particles was also tested under the same pressure, which indicated an increase in electrical resistivity ($81.8 \text{ k}\Omega \cdot \text{cm}$) after applying our insulating strategy. For the test of coercivity, it is conducted by a soft magnetic properties test system (Hunan Lianzhong Technology Co., Ltd., FE-2100SD) for the magnetic ring by winding in a closed circuit instead of powder sample (Referenced by IEC 60404-4:2000).

Then, we have also provided the hysteresis loop of powders in Supplementary Fig. 17 by the SQUID system (MPMS-XL-5, Quantum Design). A zoomed-in view of the experimental loop is shown in Fig.R1a, where the results also demonstrate that FeSiA:Sn-0.8 particles after annealing present the smallest coercivity (Fig.R1b). However, the coercivity directly extracted from the loop is larger than 20 Oe. The discrepancy of coercivity might originate from the testing system or testing method. On one hand, SQUID system has the testing range of $\pm 9 \text{ T}$, which fails to exactly control the step size of applied magnetic field when testing at low magnetic field as shown in Fig.R1. On the other hand, coercivity is an extrinsic magnetic property, which can be influenced by the stray field of powder sample. We simulated hysteresis loop of FeSiAl with and without Sn doping as shown in Fig.5b in order to confirm that the coercivity could be reduced with optimal Sn doping, which is also consistent with the experimental results in Fig.5a.

Fig.R1. (a) Hysteresis loop of powders near the coercive field in Supplementary Fig. 16a cc by SQUID. (b) Zoomed-in view of the yellow box in a.

4- While the manuscript repeatedly emphasizes the roles of coercivity, resistivity, and porosity in influencing total core loss through the reduction of hysteresis and eddy current loss components, no quantitative decomposition of these loss contributions is provided. Core loss separation, using either experimental or analytical methods, would significantly strengthen the manuscript's conclusions and help isolate the true contribution of the proposed mechanism.

Author reply: We have quantitatively separated the power loss according to the classical loss separation theory and added the corresponding results (Supplementary Fig. 14 and Table S1) in the revised paper.

The dependence of power loss on frequency for FeSiAl:Sn-0.8 SMC is measured under different flux densities (B_m), as shown in supplementary Fig.14a. According to the classical loss separation theory, power loss is composed of hysteresis loss (P_h), eddy current loss (P_e) and excess loss (P_{exc}), which can be listed below [R2-3]:

$$P_{cv} = P_h + P_e + P_{exc} = C_{hyst} B_m^\alpha f + C_{ec} B_m^2 f^2 +$$

P_{exc}

where C_{hyst} represents the hysteresis loss coefficient, α is a fitting coefficient, C_{ec} is the eddy current loss coefficient, and f is the frequency.

The corresponding P_{cv}/f is fitted in supplementary Fig.14b. It is observed that P_{cv}/f shows an obvious linear relation with f , yielding the intercept as $C_{hyst}B_m^\alpha$ and the slope as $C_{ec}B_m^2$. The corresponding R^2 is larger than 0.96, indicating good fitting results. By fitting the intercepts- B_m in supplementary Fig.14c, the coefficients of C_{hyst} and α can be derived, which is related to hysteresis loss. Meanwhile, C_{ec} can also be obtained by the slope of the fitted P_{cv}/f - f curves according to the above equation, which is related to eddy current loss. Then, the loss separation results for FeSiAl:Sn-0.8 SMC are summarized in supplementary Fig.14d. It is observed that power loss is mainly composed of hysteresis loss and eddy current loss in comparison with the negligible excess loss.

Supplementary Fig. 14. Power loss separation process. (a) Power loss of FeSiAl:Sn-0.8 SMC measured under different B_m . (b) Measured and simulation results for P_{cv}/f - f curve of FeSiAl:Sn-0.8 SMC under different B_m . (c) The intercepts of b on the vertical coordinate with black dots and the simulation curve (red line) for quasi-static hysteresis loss versus B_m . (d) The bulk loss surfaces of power loss, hysteresis loss, eddy current loss and excess loss of FeSiAl:Sn-0.8 SMC, respectively.

Using the same method, we have separated power losses of different FeSiAl:Sn SMCs. Meanwhile, the values of power loss, hysteresis loss and eddy current loss under 50 mT/100 kHz are listed in Table S1. The results indicate that both hysteresis and eddy current losses decrease with increasing Sn addition from 0 to 0.8. Meanwhile, the FeSiAl:Sn-0.8 SMC exhibits the lowest hysteresis loss (28.4 mW/cm³) attributed to the optimal Sn substitution in the FeSiAl matrix. However, when the Sn content further exceeds, the hysteresis loss sharply increases to 38.04 mW/cm³ for FeSiAl:Sn-1.0 SMC, consistent with the coercivity trend.

[R2] Bertotti, *IEEE*, (1988) 24(1): 621-630.

[R3] W. Li, H. Cai, Y. Kang, et.al, *Acta Materialia*, (2019) 167:267-274.

The revised text:

Power loss, a critical performance metric for SMCs, was characterized across all samples. The power loss for FeSiAl:Sn-0 is measured at 2573 mW/cm³ at 1 MHz under 50 mT, while all FeSiAl:Sn SMCs exhibit substantially lower power loss (**Fig. 5c**). Specifically, FeSiAl:Sn-0.8 achieves the lowest power loss of 1344 mW/cm³, primarily ascribed to suppressed hysteresis and eddy losses, which is also verified by the loss separation results in supplementary Fig.14 and Table S1. It is noteworthy that FeSiAl:Sn-1.0 displays a relatively higher power loss of 1918 mW/cm³ due to increased hysteresis loss confirmed by high coercivity, which is mainly caused by large lattice distortion ascribed to excessive Sn substitution and increased demagnetization field due to the increase of air gap induced by the porosity.

Table S1. Loss separation results of different FeSiAl:Sn SMCs under 50 mT/100 kHz.

SMCs	P_h	P_e	P_h+P_e	P_{cv}
FeSiAl:Sn-0.0	46.8	30.6	77.4	77.63
FeSiAl:Sn-0.4	32.36	24.8	57.16	57.18
FeSiAl:Sn-0.6	32.08	24.6	56.68	56.7
FeSiAl:Sn-0.8	28.4	18.3	46.7	47.3
FeSiAl:Sn-1.0	38.04	18.3	56.34	57.35

5- Although the work presents interesting experimental observations and a potentially useful approach, the novelty and impact remain limited, and several key aspects require deeper analysis and clearer validation. Based on the current scope and level of advancement, I cannot recommend the manuscript for publication in Nature Communications and it needs to be considered elsewhere with a more focus on soft magnetic materials.

Author reply: We thank the reviewer's insightful comments. We have addressed the reviewer's concerns about the novelty and experimental results. We would therefore kindly request the professional opinion of reviewer to consider this manuscript for publication in Nature Communications.

Response to Reviewer 2

Reviewer #2 (Remarks to the Author):

Comments from Ref.

The submission describes a very in depth study of a series of FeAlSi-SnO composites intended for use as low loss inductor materials for applications at 10s kHz to ~1 Mhz frequencies. The authors strive to convince the readership of originality of approach and performance.

1. The findings are indeed interesting and of value to the inductor materials community and for energy storage (inductors), power conditioning (filters), and power conversion (converters and transformers) applications at frequencies of 10s to 100s of kHz.

Author reply: We thank the reviewer's positive comments.

2. The engineering utility of these materials are clearly apparent especially in the design of data centers among other applications.

Author reply: We thank the reviewer's positive comments.

3. Although the authors claim applications to much higher frequencies. Although the definition of cut-off frequency embraced here is used by others, in practice this value is not near the operational frequency. Therefore, these alloys do not offer practical use at the frequencies cited and they provide very low permeabilities for most applications. Finally, characterization of sample set magnetization is incomplete. Authors do not attempt to put this type data in the context of comparison to other competitive products etc. This would factor into power handling and size factors.

Author reply: It is true that the application frequency range of SMCs is mostly below MHz nowadays and sharp increase in power loss has severely limited the application of SMCs at higher frequencies. However, our aim is to propose a solution that reduces the power loss of SMCs in high frequency, greatly extending the frequency range to MHz or even higher.

Herein, the effective permeability of FeSiAl:Sn-0.8 SMC declines from 60.23 (200 kHz) to 60.15 (1 MHz) with decrease only by 0.13% as given in Fig.5d. Meanwhile, the real part of permeability of FeSiAl:Sn-0.8 SMC drops only 3.9% from 1 MHz to 10 MHz (Fig.R2), which ensures the target operational frequency of hundreds of kHz or even MHz in this paper.

Fig.R2. Frequency dependence of real part of permeability (μ') of FeSiAl:Sn-0.8 SMC.

We have also added the comparison of cut-off frequency (f_c) in Table S2. Compared with other counterparts, higher cut-off frequency is achieved for our FeSiAl:Sn-0.8 SMC.

Table S2. Comparison of power loss, effective permeability and cut-off frequency (f_r) of FeSiAl:Sn-0.8 SMC with other counterparts.

SMCs	μ_e	P_{cv} (mW/cm ³)				M_s (emu/g)	f_r (MHz)
		100 mT/ 50 kHz	100 mT/ 100 kHz	50 mT/ 100 kHz	50 mT/ 1 MHz		
FeSiAl@MgO [19]	37	367	674				
Fe _{84.2} Si _{9.6} Al _{5.4} Sn _{0.8} [27]	50.4	124.4		66.3		119.2	
FeSiAl@MoO ₃ [38]	29.5			128.8		124	
FeSiAl@H ₃ PO ₄ @Na ₂ SiO ₃ [39]	50.9			133.9			
FeSiAl@SiO ₂ [40]	42.2		372			118.57	10
FeSiAl@PDA/PEI [41]	53.2	141.7	341	63.8		107.5	
FeSiAl@H ₃ PO ₄ [42]	54.2			142		122.9	
FeSiAl@hBN [43]	40.7		398	100			≥ 30
FeSiAlNi@ZnFe ₂ O ₄ [44]	49.6			506.9		145	
FeSiAl@Epoxy resin [45]	36			512		120.8	
FeSiAl@phosphate [46]	60	270					
FeSiAl@Co ₃ O ₄ [47]	56					105.5	
FeSiAl:Sn-0.8 This work	60.2	74	190	47	1344	136	250.7

4. Here are my conclusions: the work is thorough and of a very high quality; the conclusions are well-thought out and defensible; the materials and compositions are largely unoriginal (for example, the literature is replete with reports of FeAlSi alloys for inductor applications, and the use of Sn-oxides as a GB additive to disrupt eddy currents is also not new); the findings are not completely surprising nor unanticipated. Based on these comments, I believe the work does not warrant publication in Nature Comm and is better suited for other materials journals.

Author reply: Although FeSiAl is one of traditional materials, it still plays a key role in modern industry, such as integrated transformers and inductors. The breakthrough of applications at MHz range needs the supporting of novel technology. Moreover, although many papers have been published regarding FeSiAl alloys for inductor

applications, most of them concern about developing insulating coating layers to achieve only electrical isolation or heterogeneous doping in alloy matrix to achieve only low electrical resistivity. Herein, we proposed bulk/interface insulation strategy to synthesize SMCs by mutual diffusion of metal atoms during the annealing process, which not only generates an in-situ insulating layer but also improves the electrical resistivity and minimizes the coercivity.

Then, as we have responded, there is a paper about the addition of SnO₂ to Mn-Zn ferrites mentioned by the reviewer (Magnetic properties and microstructures of SnO₂ doped Mn-Zn ferrites, *Rare Metals*, 2006, 25:445). However, the paper is different from our FeSiAl:Sn SMCs, whether in the materials, compositions, methods, results or mechanisms. The main differences are listed below:

- (1) Material system: A non-metallic material of Mn_{0.57}Zn_{0.35}Fe_{2.08}O₄ was used in the corresponding paper.
- (2) Compositions: SnO₂, SiO₂ and CaO doping.
- (3) Methods: After wet ball milling for 6 h, the powders were calcined at 950 °C for 2 h, and then de-agglomerated by wet ball milling for 6 h.
- (4) The results: The highest resistivity, lowest total power loss and eddy current loss are achieved with samples doped with SnO₂-CaO-SiO₂.
- (5) Mechanisms: SnO₂ doping is expected to increase the resistivity of Mn-Zn ferrites by forming Sn⁴⁺-Fe²⁺ pairs and segregating to grain boundaries.

Herein, SnCl₂·2H₂O is used as a precursor to generate SnO by annealing in a nitrogen atmosphere on the surface of FeSiAl particles. This condition initiates a disproportionation reaction, yielding both SnO₂ and metallic Sn. Consequently, the Sn atoms diffuse inward and gradually substitute Al in the FeSiAl matrix, while outward diffusion of Al atoms forming an Al-rich layer at the particle surface. As Al and SnO₂ accumulate, an aluminothermic reaction is triggered, leading to the in-situ formation of an epitaxial Al₂O₃ insulating layer and thereby realizing the structure of FeSiAl:Sn/Al₂O₃. This Al₂O₃ insulating layer reduces inter-eddy loss attributed to electrical isolation between the FeSiAl particles. Furthermore, the substituting Sn heteroatoms can release lattice distortion during annealing treatment, resulting in

reduced coercivity and mitigating hysteresis loss. Additionally, the Sn substitutions in the FeSiAl matrix enhance its electrical resistivity, contributing to a decrease in intra-eddy loss. Our work not only elucidates a novel methodology for enhancing high-frequency performance in FeSiAl SMCs but also offers insights applicable to the development of other material systems, potentially guided by the principles of heteroatom substitution and atom diffusion.

We would therefore kindly request the professional opinion of reviewer to consider this manuscript for publication in Nature Communications.

Response to Reviewer 3

Reviewer #3 (Remarks to the Author):

The authors have addressed my concerns in their response and revised manuscript. I recommended it for publication in Nature Comm. as is.

Response to Reviewer 4

Reviewer #4 (Remarks to the Author):

The authors report a promising strategy for simultaneously reducing hysteresis and dynamic losses (including both intra- and inter-particle eddy currents) in a FeSiAl-based soft magnetic composite (SMC). To my knowledge, numerous studies on FeSiAl have utilized Al₂O₃ as an insulating coating, including the paper from the same research group cited by Reviewer 1. Furthermore, the strategy of doping with heteroatoms to modify the crystal lattice, such as with Ti or Sn, has been previously explored. In this regard, while the authors draw comparisons to prior work on Sn doping, this research (doi.org/10.1016/j.jmrt.2022.04.078) is not discussed in the introduction. To better contextualize and highlight the specific advancements of the current manuscript, a more thorough discussion of this reference in the Introduction would be highly beneficial.

The novelty here lies in combining these two strategies, which have previously only been used in isolation. The clever choice of precursor has allowed to generate an in-situ insulating layer while improving the electrical resistivity and minimizing the coercivity of the ferromagnetic particle, yielding results that are not only interesting from an application standpoint but also inspirational for developing new concepts in this class of materials.

Author reply: We have added thorough discussion of the corresponding research in the Introduction and made them highlighted in the revised paper.

The revised text:

Consequently, implementing heterogeneous element doping in master alloys emerges as a promising strategy to improve electrical resistivity and inhibit power loss. For instance, Wang et al. demonstrated an enhanced electrical resistivity by doping Ti in Fe-6.5Si alloy²⁴. Liu prepared Fe_{85-x}Si_{9.6}Al_{5.4}Sn_x powders by induction melting and gas atomization and the corresponding SMC exhibited the lowest core loss of 124.4 mW/cm³ ascribed to the addition of Sn in Fe sites²⁵. Similar findings have been reported in Fe_{85-x}Si_{9.6}Al_{5.4}Ti_x alloy, wherein power loss was effectively reduced to

approximately 140.6 mW/cm³ under the same conditions ²⁶. Nevertheless, unexpected diffusion and segregation happen for the doping alloy during following gas atomization and annealing processes, leading to lattice distortion and influencing magneto-crystalline anisotropy ²⁷, which in turn degrades coercivity (H_c) and hysteresis loss in SMCs.

Regarding the manuscript, my concerns are as follows:

1. Reviewer 1 questioned why the Sn-1.0 sample, despite having the highest resistivity, exhibits higher power loss than the Sn-0.8 sample. The authors argue this is due to a trade-off: beyond the optimum of Sn-0.8, the increase in hysteresis loss (from higher coercivity due to excessive lattice distortion) outweighs the benefit of increased resistivity. While this explanation is plausible, it overlooks a critical confounding variable: porosity. The authors' own data show that the density of the SMCs decreases with increasing Sn content (6.02 g/cm³ for Sn-0, 6.0 g/cm³ for Sn-0.8, and 5.98 g/cm³ for Sn-1.0). Lower density implies higher porosity, which acts as an effective air gap, increasing the demagnetization field, increasing hysteresis losses, and reducing the effective permeability (μ_e). This effect is evident in Fig. 5d, where μ_e for the Sn-1.0 sample is the lowest. The authors attribute the drop in μ_e solely to "magnetic dilution," but the density data suggest porosity is a contributing factor that has not been discussed. The authors should expand their discussion on the Sn-1.0 sample's performance to include the likely contribution of its lower density and higher porosity.

Author reply: We have expanded discussion on the Sn-1.0 sample's performance in the revised paper.

The revised text:

Power loss, a critical performance metric for SMCs, was characterized across all samples. The power loss for FeSiAl:Sn-0 is measured at 2573 mW/cm³ at 1 MHz under 50 mT, while all FeSiAl:Sn SMCs exhibit substantially lower power loss (**Fig. 5c**). Specifically, FeSiAl:Sn-0.8 achieves the lowest power loss of 1344 mW/cm³, primarily ascribed to suppressed hysteresis and eddy losses, which is also verified by the loss separation results in supplementary Fig. 14 and Table S1. It is noteworthy that

FeSiAl:Sn-1.0 displays a relatively higher power loss of 1918 mW/cm³ due to increased hysteresis loss confirmed by high coercivity, which is mainly caused by large lattice distortion ascribed to excessive Sn substitution and increased demagnetization field due to the increase of air gap induced by the porosity. Meanwhile, FeSiAl:Sn-0.8 exhibits the highest quality factor (Q) as indicated in supplementary Fig. 15. It should be noted that the reduced inter and intra-eddy losses of FeSiAl:Sn/Al₂O₃ structure are also verified by the inhibited current density cloud and corresponding vector maps in supplementary Fig. 16.

Additionally, FeSiAl:Sn-0.8 SMC demonstrates a steady effective permeability of 60 across the frequency range of 0 to 1 MHz (**Fig. 5d**). The effective permeability decreases with increasing SnCl₂ content, following the same trend as density. In addition to magnetic dilution due to the addition of SnCl₂ to the SMC, higher porosity evidenced by lower density also introduces air gaps and enlarges demagnetization field in the SMC, thereby reducing the effective permeability of FeSiAl:Sn-1.0 SMC. Meanwhile, the saturation magnetizations of FeSiAl:Sn particles decrease from 144 to 136 emu/g (supplementary Fig. 17a).

2. The "Material preparation" section lacks sufficient detail to ensure the work's reproducibility. While some points were addressed in response to other reviewers, I suggest the authors clarify the following:

(a) The mass of the FeSiAl particles mixed with the 0.8 g of tin(II) chloride dihydrate.

Author reply: We have provided the mass of FeSiAl in the revised paper.

The revised text:

The above solution was evenly dispersed into 500 g FeSiAl particles (TIJO, 400 mesh with D10=5.58 μm, D50=15.64 μm, D90=33.47 μm. FeSi_{9.1}Al_{5.4} (wt.%) with apparent density:3.83 g/cm³) and mechanically stirred for 10 minutes, so that the solution was evenly coated on the surface of FeSiAl.

(b) The volume of ethanol used to dissolve the tin chloride and any additional parameters (e.g., stirring speed, temperature).

Author reply: We have provided the volume of ethanol, stirring speed and temperature in the revised paper.

The revised text:

0.8 g Tin chloride dihydrate ($\text{SnCl}_2 \cdot 2\text{H}_2\text{O}$) was dissolved in 20 ml absolute ethanol, and the above solution was magnetically stirred for 30 minutes with 650 rpm at room temperature to obtain a transparent solution.

(c) The method used to mix the FeSiAl particles with the tin chloride solution (e.g., mechanical, magnetic) and the associated parameters (e.g., speed, temperature, duration).

Author reply: We have provided the corresponding method in the revised paper.

The revised text:

Then, the above solution was evenly dispersed into 500 g FeSiAl particles (TIJO, 400 mesh with $D_{10}=5.58 \mu\text{m}$, $D_{50}=15.64 \mu\text{m}$, $D_{90}=33.47 \mu\text{m}$. $\text{FeSi}_{9.1}\text{Al}_{5.4}$ (wt.%) with apparent density: 3.83 g/cm^3) and mechanically stirred for 10 minutes, so that the solution was evenly coated on the surface of FeSiAl. Then, the mixed composite was heated at $80 \text{ }^\circ\text{C}$ for 20 minutes (termed as FeSiAl:Sn particles).

(d) The drying method and parameters (e.g., temperature, duration, atmosphere) used after mixing, if any.

Author reply: We have provided the drying method in the revised paper.

The revised text:

Materials preparation. 0.8 g Tin chloride dihydrate ($\text{SnCl}_2 \cdot 2\text{H}_2\text{O}$) was dissolved in 20 ml absolute ethanol, and the above solution was magnetically stirred for 30 minutes with 650 rpm at room temperature to obtain a transparent solution. Then, the above solution was evenly dispersed into 500 g FeSiAl particles (TIJO, 400 mesh with $D_{10}=5.58 \mu\text{m}$, $D_{50}=15.64 \mu\text{m}$, $D_{90}=33.47 \mu\text{m}$. $\text{FeSi}_{9.1}\text{Al}_{5.4}$ (wt.%) with apparent density: 3.83 g/cm^3) and mechanically stirred for 10 minutes, so that the solution was evenly coated on the surface of FeSiAl. Then, the mixed composite was heated at $80 \text{ }^\circ\text{C}$ for 20 minutes (termed as FeSiAl:Sn particles).

(e) The mixing procedure for the FeSiAl particles with the 5 g of silicone resin and 5 g of zinc stearate, including the mass of the FeSiAl powder used.

Author reply: We have provided the mixing procedure in the revised paper.

The revised text:

The above FeSiAl:Sn particles were mechanically mixed with 5 g silicone resin dissolved in 15 ml acetone and dried at 80 °C for 10 minutes. After adding 5 g zinc stearate, the powder was molded into toroidal cores (12.7 mm×7.6 mm×3.2 mm) under 1800 MPa.

(f) What happens to the silicone resin after the annealing process? Was it completely burned off?

Author reply: After the annealing process in N₂, the silicone resin completely decomposes to the amorphous silicon oxycarbide (SiOC) structure. The molecular backbone of silicone resin consists of silicon-oxygen (Si-O) bonds, with methyl groups (-CH₃) and phenyl groups (-C₆H₅) attached to the side chains. Silicone resin is not resistant to high temperatures, and its thermal decomposition temperature is generally less than 500 °C. During the annealing treatment, C-H bonds break and organic groups are completely eliminated at approximately 600°C as evidence by TGA (Fig.R3a). After annealing treatment at 700 °C for 60 minutes in N₂, silicone resin transforms from an organic polymer to the SiOC structure with absence of Si-CH₃ and C-H absorption peaks (Figs.R3b-d) ^[R4-5]. Meanwhile, the amorphous SiOC agglomerates at the particle interface as indicated in the EDS mapping images in Fig.R3d.

Fig.R3. (a) TGA curve of silicone resin. (b) FTIR spectra of silicone resin and silicone resin after annealing. (c) SAED pattern of silicone resin after annealing treatment at 700 °C for 1h. (d) HAADF image and corresponding EDS mapping of FeSiAl:Sn-0.8 SMC.

[R4] B Kumar, RM Prasad, *Sensors and Actuators B Chemical*, (2023) 379:133226.

[R5] Kwang-ManLee, Young Kim Chang, et.al, *Journal-Korean Physical Society*, (2011) 59:3074.

(g) The authors only mentioned that the particles (I am assuming mean size) are 400 mesh. However, Supplementary Fig. 15 reveals a quite large distribution size for the particles. Please, inform the d10, d50, and d90 of the distribution, as it can largely affect the intra-eddy losses.

Author reply: For FeSiAl powder, the D10=5.58 μm, D50=15.64 μm and D90=33.47 μm have been provided in the Methods.

3. Part of the caption for Figure 5 refers to measurements at 100 mT, but these data are not shown in the figure itself. The referenced data appear in Supplementary Figure S17, which lacks a caption. Please correct this discrepancy to improve clarity for the reader.

Author reply: We have corrected the discrepancy to improve clarity in Supplementary Fig. 17 according to the reviewer's comments.

Supplementary Fig. 17. Comparison of power loss of our FeSiAl:Sn-0.8 SMC with other counterparts measured at 100 mT.

=====**Authors' responses to the reviewers' comments**=====

Response to Reviewer 1

Reviewer #1 (Remarks to the Author):

The authors have addressed all the questions raised earlier. I would like to recommend the revised manuscript for publication as it is.

Author reply: We thank the reviewer's positive comments.

Response to Reviewer 4

Reviewer #4 (Remarks to the Author):

The authors have addressed my concerns in their response and revised manuscript. I recommended it for publication in Nature Comm. as is.

Author reply: We thank the reviewer's positive comments.